# Global census of the significance of giant mesopelagic protists to the marine carbon and silicon cycles

Manon Laget [1] ✉, Laetitia Drago[2], Thelma Panaïotis [2], Rainer Kiko[3], Lars Stemmann [2], Andreas Rogge [4], Natalia Llopis-Monferrer[5,6], Aude Leynaert [7], Jean-Olivier Irisson[2] & Tristan Biard [1]

Thriving in both epipelagic and mesopelagic layers, Rhizaria are biomineralizing protists, mixotrophs or flux-feeders, often reaching gigantic sizes. In situ imaging showed their contribution to oceanic carbon stock, but left their contribution to element cycling unquantified. Here, we compile a global dataset of 167,551 Underwater Vision Profiler 5 Rhizaria images, and apply machine learning models to predict their organic carbon and biogenic silica biomasses in the uppermost 1000 m. We estimate that Rhizaria represent up to 1.7% of mesozooplankton carbon biomass in the top 500 m. Rhizaria biomass, dominated by Phaeodaria, is more than twice as high in the mesopelagic than in the epipelagic layer. Globally, the carbon demand of mesopelagic, flux-feeding Phaeodaria reaches 0.46 Pg C y$^{-1}$, representing 3.8 to 9.2% of gravitational carbon export. Furthermore, we show that Rhizaria are a unique source of biogenic silica production in the mesopelagic layer, where no other silicifiers are present. Our global census further highlights the importance of Rhizaria for ocean biogeochemistry.

Life in the surface ocean produces biogenic material that is constantly exported to depth, fueling deep-sea ecosystems with nutrients and minerals. Exported particulate organic carbon (POC) is the basis of the biological carbon pump (BCP), a key process in regulating atmospheric $CO_2$ levels[1]. This export results from various pathways, including transport by gravitational settling of particles, vertically migrating organisms, and lateral transport[2]. Just below the epipelagic layer where light becomes insufficient for photosynthesis (typically around 200 m), the mesopelagic layer receives a rain of organic material punctuated by episodic inputs from the surface ocean[3]. This mesopelagic realm is a vast transition zone where ecological interactions determine the fate and amount of material that will ultimately reach the deep ocean.

Because of inherent sampling constraints, our knowledge of stocks and processes is scarcer in the deep than in the upper ocean, leaving severe uncertainties in the response of the BCP to global changes[4]. Settling particles are sources of food for heterotrophic organisms, which consume ~90% of POC exported from surface waters before it reaches 1000 m depth[3]. Therefore, conducting a census of the mesopelagic biota and quantifying its contribution to biogeochemical cycling in regard to its trophic role is essential to understand the fate of sinking material[3,5]. Most previous studies on mesopelagic biota focused on morphologically robust metazoan taxa, which are more easily accessible using nets or active acoustics[6,7]. In contrast, the role of unicellular zooplankton such as

[1]LOG, Laboratoire d'Océanologie et de Géosciences, Univ. Littoral Côte d'Opale, Univ. Lille, CNRS, IRD, UMR 8187, Wimereux, France. [2]Sorbonne Université, CNRS, Laboratoire d'Océanographie de Villefranche (LOV), Villefranche-sur-Mer, France. [3]GEOMAR Helmholtz Center for Ocean Research Kiel, Kiel, Germany. [4]Section Benthic Ecology, Alfred Wegener Institute Helmholtz Center for Polar and Marine Research (AWI), Bremerhaven, Germany. [5]Monterey Bay Aquarium Research Institute, Moss Landing, CA, USA. [6]Sorbonne University, CNRS, UMR7144 Adaptation and Diversity in Marine Environment (AD2M) Laboratory, Ecology of Marine Plankton team, Station Biologique de Roscoff, Roscoff, France. [7]Université de Brest, CNRS, IRD, Ifremer, LEMAR, Plouzané, France. ✉e-mail: manon.laget@protonmail.com

Rhizaria, known to be abundant in this layer[8–10], has been overlooked.

Rhizaria are a 515-Ma ancient eukaryotic lineage[11], among which planktonic taxa populate modern oceans from the surface to the abyss[8,9,12] and from equatorial to polar waters[8–10]. Their size range spans from a few μm to several mm, with some taxa being able to form colonies of up to one meter[9]. Planktonic Rhizaria include Phaeodaria, Radiolaria, and Foraminifera[8,9]. Phaeodaria and Radiolaria (including the orders Acantharia, Orodaria, and Collodaria) mostly biomineralize siliceous skeletons, while many Collodaria are naked and Acantharia build strontium sulfate skeletons[8,9]. These groups are often constrained to a narrow depth range[12], according to their trophic mode and ecological niche.

Many Radiolaria and Foraminifera are mixotrophs, harboring photosynthetic algal endosymbionts which sustain energetic requirements of the host cell[13]. They contribute to atmospheric $CO_2$ uptake and can thrive in the epipelagic layer of oligotrophic oceans, where organic food resources are scarce[9]. Phaeodaria, on the other hand, are strictly heterotrophic and mostly found in the mesopelagic layer, where they feed upon sinking particles[14,15]. Many large planktonic Rhizaria have adapted their lifestyle to food-depleted environments through a low cellular carbon density[16]. In contrast, siliceous Rhizaria are the most silicified pelagic organisms known to date[17,18]. As they are abundant down to the bathypelagic ocean, they contribute to silica uptake where no other planktonic organism takes up dissolved silica[18]. Siliceous skeletons of Phaeodaria can act as ballasting minerals upon organisms' death and increase the settling velocity of incorporated and attached POC towards the deep ocean. However, these organisms are flux-feeders, i.e., they feed on sinking particles rather than suspended ones[14,15]. As a result, they can attenuate a substantial amount of sinking flux especially in the upper mesopelagic, where they are most abundant[19].

Because traditional sampling techniques damage fragile representatives especially of large Rhizaria, these organisms have often been neglected in biogeochemical studies[20]. Nevertheless, in situ imaging tools have revealed their substantial contribution to elemental stocks[20,21] and fluxes[17], as well as their role as gatekeepers of the BCP[19,22]. Yet of interest for biogeochemical modeling and our understanding of the biological pump, their capacity to contribute to flux attenuation has never been assessed to a larger extent.

In this study, we assess the global organic carbon and silica biomasses of large Rhizaria as well as the role of large Phaeodaria in mediating the fluxes of these elements. While we revise downward the total carbon biomass of Rhizaria, we show that these protists are widely distributed at high latitudes, where their abundance was previously shown to decline[20]. Furthermore, we highlight the importance of large Phaeodaria in the mesopelagic layer and show that they can attenuate 3.8–9.2% of gravitational carbon export. We find that these silicifying protists co-dominate the silicon cycle, along with diatoms and sponges. In particular, they constitute a unique source of biogenic silica (bSi) production and stock in the mesopelagic layer, where no other silicifiers are found. With expected changes in oceanic conditions and in global carbon export, we discuss the role of these protists in future oceans.

## Results and discussion

### Collecting Rhizaria images and modeling their distribution

Here, we present a global dataset of large Rhizaria distribution and abundance collected in situ throughout all oceans between 2008 and 2021. The dataset consists of 4252 vertical profiles acquired with the Underwater Vision Profiler 5[23] (UVP5), of which 1959 extend down to 1000 m depth (Fig. 1 and Supplementary Table 1). The UVP5 recorded 74,157 images in the epipelagic layer (0–200 m) and 93,394 images in the mesopelagic layer (200–1000 m), which have been all manually validated. With Equivalent Spherical Diameters (ESD) ranging from 0.6 to 20 mm (therefore excluding more abundant smaller Rhizaria; Supplementary Table 2 and Supplementary Fig. 1), Rhizaria images cover 18 taxonomic categories[12] included in Radiolaria, Foraminifera, Phaeodaria and other Rhizaria (Fig. 2 and Supplementary Table 2). We apply the most recent allometric volume-to-elemental content relationships[16,18] to obtain carbon content of all Rhizaria groups, as well as the silica content for Phaeodaria only. Indeed, all Phaeodaria are known as silicifying, while silicified Collodaria cannot be distinguished from naked ones in UVP5 images. Besides their contribution to

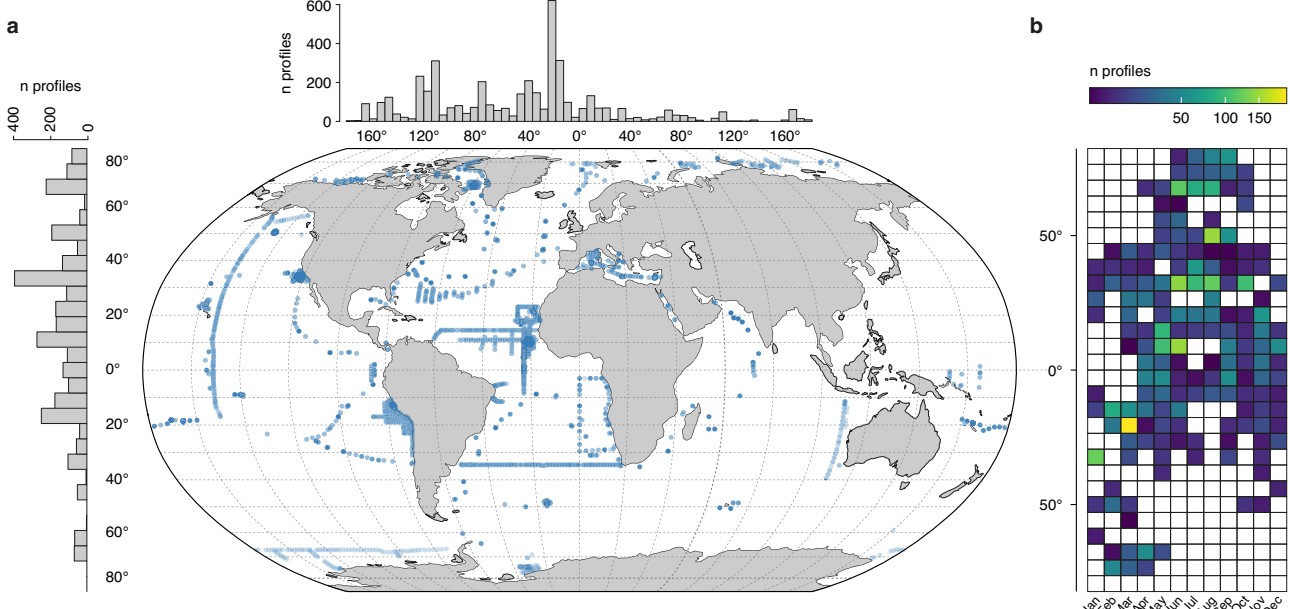

**Fig. 1 | Spatial and temporal coverage of the 4252 vertical in situ profiles.**
**a** Geographic location of Underwater Vision Profiler 5 profiles used in this study. Left and top histograms show the latitudinal and longitudinal distribution of profiles along 3° bins. **b** Seasonal distribution of sampling points according to latitude. Squares are colored proportionally to the sampling effort. Details about sampling are summarized in Supplementary Table 1. The map was created using the R software version 4.0.3 (ref. 63).

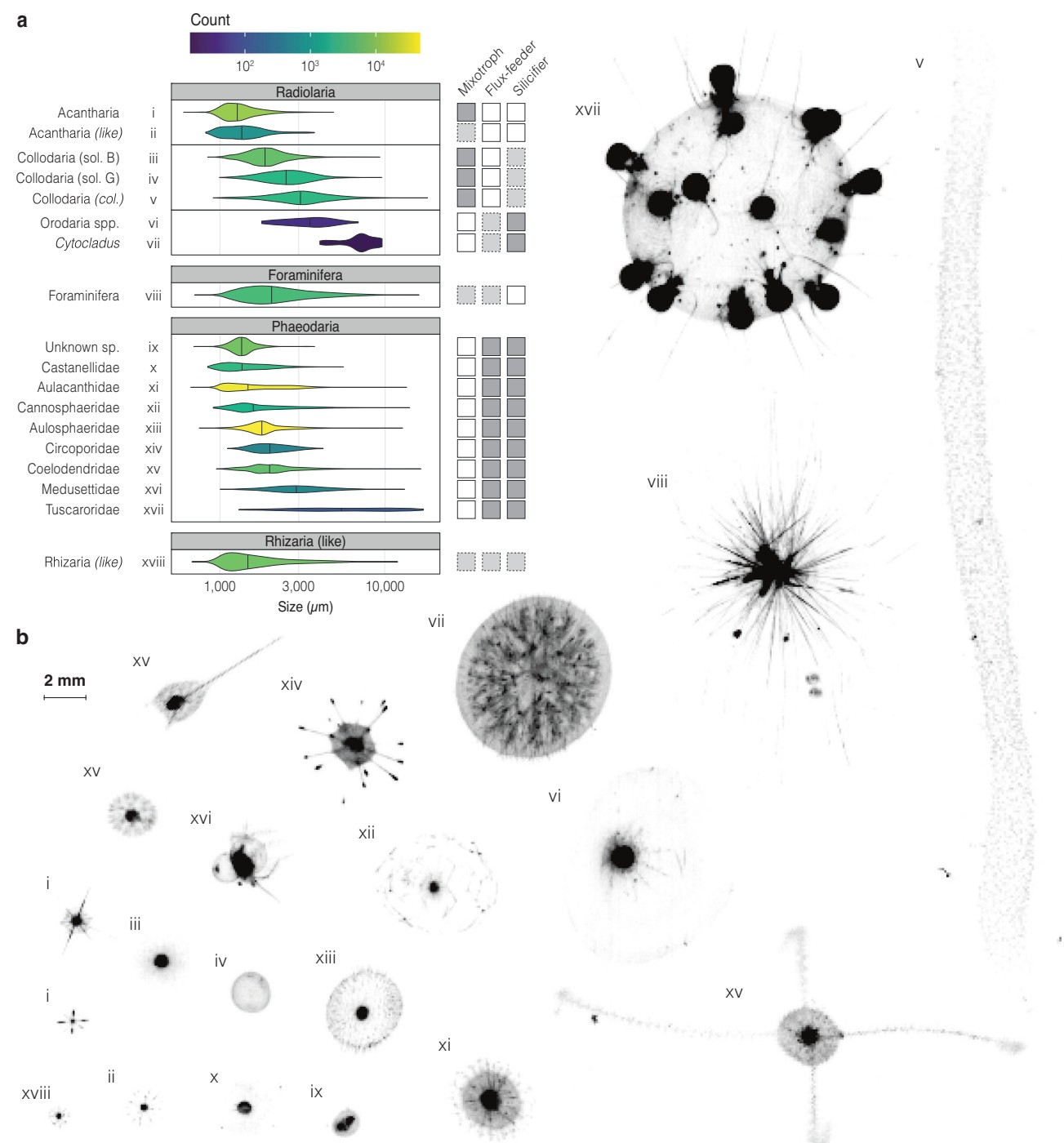

**Fig. 2 | Overview of sampled Rhizaria specimens. a** Equivalent spherical diameter distribution, counts, and characteristics of each taxon considered. Characteristic mixotrophic, flux-feeding (feeding on sinking rather than suspended particles), and silicifying (the capacity to build a silica skeleton) lifestyles are indicated on the right-hand side (dark gray indicates a proven attribute, light gray an evident attribute for at least parts of the species within this taxon, and white its proven absence). **b** Example images for analyzed Rhizaria taxa numbered according to (**a**) (all images are on the same scale).

element stocks, we further investigate their role in biogeochemical processes by estimating Phaeodaria carbon demand[22] and bSi production[18].

Associating each of the 167,551 Rhizaria images with a set of environmental variables, we quantify these biomasses and processes on a global 1° × 1° grid by using boosted regression trees, with both random and spatial cross-validation (CV) for model assessment. We used spatial CV to account for the fact that spatial data are not independent and to assess the performance of our models on new environmental conditions. For each model and CV method, we provide a $R^2$ value (Supplementary Tables 3 and 4), calculated as the squared Pearson correlation coefficient between the observed biomass and the mean predicted biomass (see "Methods").

**Global Rhizaria biomass from the epipelagic to the mesopelagic**

Our models predict the global carbon biomass of large Rhizaria (>0.6 mm) within the upper 1000 m to be 0.012 Pg C (0.11–0.13 Pg C; see "Methods" for uncertainty estimation; Supplementary Table 3) and 0.007 Pg C within the top 500 m, obtained by integrating biomass within the epipelagic layer and the top 300 m of the mesopelagic layer.

This is about one to two orders of magnitude lower than prior estimates (0.204 Pg C and 0.061 Pg C in the top 500 m)[20,21]. In contrast to these studies, our estimates rely on dedicated volume-to-elemental content allometric relationships established from carbon and volume measurements on living specimens[16], which showed lower carbon densities of large Rhizaria and the inadequacy of previous conversion factors. Yet, large discrepancies can be observed in model assessment ($R^2$ metrics; Supplementary Table 3) depending on the use of random or spatial CV. By using a spatial CV, we purposely excluded parts of the observed environmental conditions, leading to highly conservative $R^2$ values. Low observed $R^2$ values when using spatial CV show the difficulty for our models to predict on new data, but it should be kept in mind that the final model is trained on the entire dataset and therefore includes a larger range of environmental conditions.

Based on total mesozooplankton biomass estimates[21], we now assess the contribution of large Rhizaria to total mesozooplankton biomass in the top 500 m to be 1.7%. Although we revise downward their contribution to biomass, the statement that these low carbon density organisms could contribute globally to 31% of mesozooplankton abundance from UVP datasets[20] still holds true.

All taxa considered, large Rhizaria are distributed worldwide in both the epipelagic and the mesopelagic layers (Fig. 3a–c). By extending our spatial coverage poleward compared to other studies, we reveal the prevalence of these organisms at high latitudes, where their biomass was previously shown to be lower[10,20,24]. However, it should be kept in mind that high-latitude sampling mostly occurred in boreal and austral summers, likely during periods of highest biomass. Most importantly, the worldwide carbon concentration of large Rhizaria is of the same order of magnitude between the epipelagic and the mesopelagic layer in the tropical ocean and at high latitudes (Fig. 3a, b). Integrated biomass values are similar between layers around 20°N and S, but get consistently higher in the mesopelagic near the equator and above 30°N and S (Fig. 3c). The two peaks at 50°N and 60°S follow recent biomass pattern delineations in subpolar regions and around the equator[21,24], likely due to the presence of fronts at these high latitudes. Nonetheless, these global patterns hide taxon-driven differences.

## Contrasting patterns between mixotrophic and heterotrophic taxa

Given their different trophic modes and lifestyles, we observe distinct patterns between mixotrophic and heterotrophic Rhizaria groups (Fig. 3d and Supplementary Fig. 2). Globally, Collodaria, Acantharia, and, to a lesser extent, Foraminifera, make up most of the Rhizaria biomass in the epipelagic layer of inter-tropical regions (45°N–45°S), particularly within subtropical oligotrophic gyres (Supplementary Fig. 3). This pattern is expected given the mixotrophic nature of these radiolarian orders[13,20,25]. Through nutrient retention, mixotrophs can enhance primary production[13], besides shortcutting energy pathways to higher trophic levels[13,20]. As these environments are expected to expand as a consequence of global changes[26], the importance of these protists in ecosystem functioning is likely to increase in warmer and more oligotrophic oceans. More investigations about their role in $CO_2$ uptake and carbon export is yet needed to provide more detailed predictions about their role in future oceans.

In contrast to mixotrophic taxa, Phaeodaria dominate Rhizaria biomasses in the mesopelagic layer (Fig. 3d and Supplementary Fig. 2). Unlike Radiolaria which prey upon various organisms ranging from bacteria to small Metazoa[25], these nonmotile organisms are floating in particle-rich zones, where they intercept aggregates by extending cytoplasmic strands (i.e., pseudopodia)[8]. Often thought to be restricted to deep waters[8], our observations reveal important epipelagic biomasses of Phaeodaria in several high-latitude areas, where they could feed on the sinking of large particles. This is in agreement with previous observations in the Southern Ocean[27–29], in particular the

Weddell Sea[29], the North Pacific[20,30], but also the Sea of Japan[31]. Globally, their total carbon biomass is tenfold higher in the mesopelagic than in the epipelagic zone (Fig. 3d and Supplementary Table 3), with an overall contribution of 81% to total Rhizaria biomass. The prevalence of Phaeodaria in deep waters and in cold high-latitude areas can be explained by their adaptation to cold-water environments, due to their gigantic size and low cellular carbon density. Despite their important abundance and biovolume[20] in many regions and depths, a result of their low carbon density is that their average contribution to total mesozooplankton carbon biomass is low (0.9% in the top 500 m). However, it is also highly variable in different ocean regions: their proportion on the total C biomass ranges from 2.7 to 13.7% between 150 and 1000 m in the North Pacific[30] up to 22.3% between 250 and 3000 m in the Sea of Japan[31]. Since these abundant mesopelagic detritivores are flux-feeders[14,15], we further investigated their metabolic requirements and potential impact on carbon fluxes in the mesopelagic layer.

## Role of mesopelagic Phaeodaria in carbon flux attenuation

We estimated the individual carbon demand of mesopelagic Phaeodaria based on their carbon content and the temperature at their depth, using a range of turnover time values (the time it takes for one individual to be replaced) and a gross growth efficiency of 40% (see "Methods"). When predicted at global scale, we estimated an annual carbon demand of 0.46 Pg C y$^{-1}$ by mesopelagic Phaeodaria alone (Supplementary Table 4). Scaled to recent global gravitational POC export estimates, ranging from 5 to 12 Pg C y$^{-1}$ (see ref. 32 and references therein), they would intercept between 3.8 and 9.2% of the gravitational POC flux exported out of the euphotic zone (Fig. 3e). By modulating their turnover times over the full span of observed values[22] (see "Methods"), the global flux attenuation by Phaeodaria could range from 2.4–5.8% to 5.0–12.0%. Considering a gross growth efficiency of 20% (see "Methods"), Phaeodaria-mediated attenuation could double. Attenuation from these large heterotrophic protists has never been taken into account in previous assessments of zooplankton carbon flux attenuation, which was thought to be driven by Metazoa[3,6,7]. Integrating this number into the global carbon budget is therefore necessary to refine biogeochemical models and to improve predictions of the future ocean state.

Maximum potential Phaeodaria-driven attenuation is found in the Southern Ocean, where it ranges between 11.2 and 23.4% of the gravitational POC export (0.62–1.3 Pg C y$^{-1}$, refs. 33, 34). In contrast, it approximates 3.8–6.7% in equatorial and upwelling areas[32,34] (Fig. 3e), where large Phaeodaria are also abundant. These observations align with previous findings showing more important zooplankton carbon demands in high latitudes and productive areas[7]. Their daily integrated carbon demand averages $3.9 \pm 3.4$ mg C m$^{-2}$ d$^{-1}$ worldwide (Fig. 3f and Supplementary Table 5), with the highest values in the subarctic Pacific ($13.0 \pm 6.5$ mg C m$^{-2}$ d$^{-1}$; Fig. 3f and Supplementary Table 5). In subtropical gyres, where they are least abundant, they could account for 18.9% of the total zooplankton carbon demand ($3.4 \pm 3.1$ mg C m$^{-2}$ d$^{-1}$ from 17.9 mg C m$^{-2}$ d$^{-1}$ reported for the North Pacific Subtropical Gyre[35]). In contrast, in the subarctic Pacific where their demand is maximum, their contribution to total zooplankton demand (133.1 mg C m$^{-2}$ d$^{-1}$, ref. 7) drops to 9.7%, as metazoan zooplankton may outcompete them for food.

As Phaeodaria are known to consume preferably sinking particles, rather than suspended ones[14,15], they can exert a substantial influence on rapidly sinking particles, which are expected to be preferentially transferred to the deep ocean due to their high sinking speed. In the California Current, the Phaeodaria family Aulosphaeridae alone can be responsible for an average 10% of total flux attenuation at the depth of their maximum abundance[19] in the upper mesopelagic zone where also food supply is highest. Due to the fact that a substantial portion of Phaeodaria also feed in the lower epipelagic above 200 m[12,19]

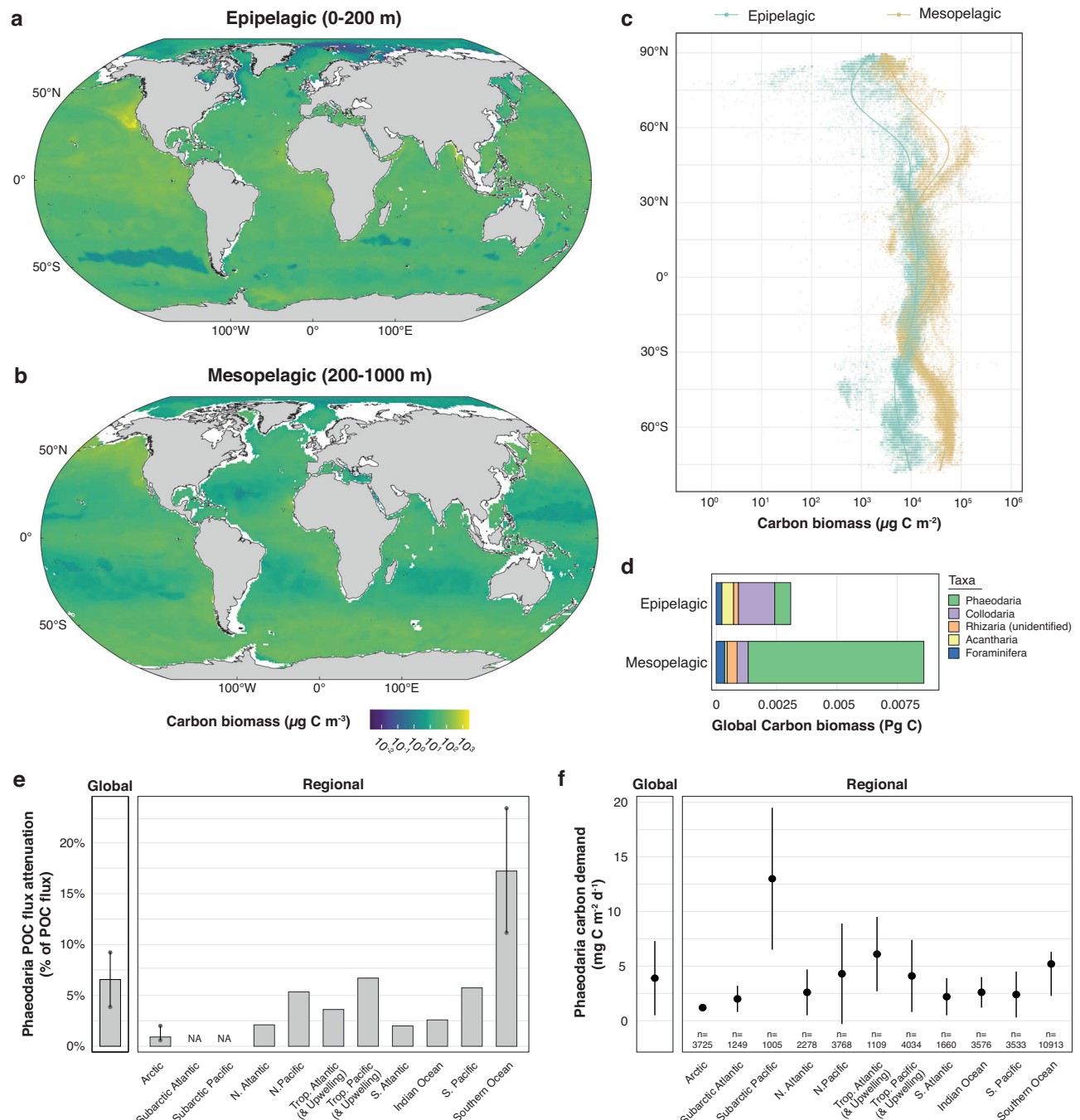

**Fig. 3 | Predicted global organic carbon biomass in planktonic Rhizaria, their impact on carbon flux attenuation, and their carbon demand. a, b** Maps of the predicted average 1° × 1° carbon concentration in the epipelagic (0–200 m; **a**) and mesopelagic layer (200–1000 m; **b**). **c** Integrated carbon biomass as a function of latitude for both layers. Regression curves were derived using Generalized Additive Models. Note the logarithmic scaling for carbon biomass. **d** Total carbon biomass for Rhizaria groups in both layers. Only groups whose model $R^2$ calculated by random cross-validation is >0.05 are shown (see also Supplementary Table 3). **e** Global and regional contribution of large mesopelagic Phaeodaria to gravitational particulate organic carbon (POC) flux attenuation, based on the ratio of their annual carbon demand to the respective median annual carbon export (i.e.,

transport out of the euphotic zone) as reported elsewhere[32–34] and summarized in Supplementary Table 5. Bars show the ratio calculated with the single export value available or the median value when more than one export value was available. In the latter case, dots show export values used to calculate the median. Error bars are represented according to the range of observed export values. No POC export measures were available for the subarctic Pacific and Atlantic. **f** Global and regional integrated daily carbon demand of large mesopelagic Phaeodaria. Dots show mean values, error bars show regional mean ± standard deviation as presented in Supplementary Table 5. The number of 1° × 1° cells used to derive statistics are displayed below for each region. **a, b** Maps were created using the R software version 4.0.3 (ref. 63).

(Supplementary Fig. 4), our estimates are likely conservative as we only estimated their carbon demand for depths between 200 and 1000 m. Ingested material is used for Phaeodaria growth but is ultimately processed into mini-pellets, ejected into the water column, and thought to play an important role in carbon export[36]. Indeed, mini-

pellet abundance can be almost five orders of magnitude higher than that of krill fecal pellets in the Weddell Sea[29]. This leaves an open question regarding whether mesopelagic Phaeodaria ultimately slow down or accelerate downward fluxes. To answer it, the nature of the exported material—either fecal pellets or aggregates[37,38]—and the

associated differences in sinking velocities and carbon contents, as well as the resulting effects on global export must be further explored. As Phaeodaria aggregates can be observed with in situ imaging[38], this tool can be used to estimate Phaeodaria-mediated fluxes. Their silicified skeleton, often included in aggregates[37], is likely increasing their sinking velocities; thus, we further investigated the role of Phaeodaria in the silicon cycle and potential impacts for the carbon cycle.

## Biogenic silica stocks and production mediated by Phaeodaria

Globally, our models predict that Phaeodaria account for 4.25 (3.8–4.7) Tg of bSi standing stocks in the upper 1000 m of the oceans, among which 0.34 (0.28–0.39) Tg are in the epipelagic and 3.91 (3.51–4.31) Tg in the mesopelagic layer (Fig. 4 and Supplementary Table 4). The mean integrated value for the epipelagic layer is $1.1 \pm 3.3$ mg Si m$^{-2}$ (range 0–204.4 mg Si m$^{-2}$; Fig. 4c), which is in the range of values measured on small Rhizaria, not within the size spectrum of the UVP5[18]. The whole population of Rhizaria would thereby account for 7–18% of the total integrated bSi pool in the epipelagic layer[39] while most of the remaining stock is being attributed to diatoms[40]. In the mesopelagic layer, the mean integrated value is $11.6 \pm 12.1$ mg Si m$^{-2}$ (range 0–378.3 mg Si m$^{-2}$; Fig. 4c) and would account for -0–15% of the total bSi integrated pool, based on average estimates from other studies[41,42].

We further quantify the role of Phaeodaria in the silicon cycle by estimating total bSi production (or dissolved silica uptake). Average global annual bSi production by Phaeodaria is 0.70 Tg Si y$^{-1}$ (range 0.22–2.33) in the epipelagic and 3.96 Tg Si y$^{-1}$ (range 1.25–13.1) in the

mesopelagic zone (Supplementary Table 4) while daily integrated production rates averages $0.005 \pm 0.016$ mg Si m$^{-2}$ d$^{-1}$ in the epipelagic and $0.030 \pm 0.031$ in the mesopelagic layer. In the Southern Ocean, the mean production rate within the top 200 m ($0.006 \pm 0.004$ mg Si m$^{-2}$ d$^{-1}$) is about three orders of magnitude lower compared to previous estimates that included small Rhizaria only[28]. Although biomasses and production rates in the epipelagic layer are seemingly low compared to diatoms[28,40], the overwhelming majority of Phaeodaria is located in the mesopelagic zone (Fig. 4). Our results complete previous bSi production rates estimated in the global ocean[18] from plankton net samples. While this first estimate (56–1600 Tg Si y$^{-1}$) was based mainly on small organisms, our 4.66 Tg Si y$^{-1}$ estimate focuses on the largest specimens (>600 μm) and is therefore difficult to compare. With the most recent estimates[40] only focusing on diatom- and sponge-mediated bulk bSi production in the sunlit epipelagic and benthic environments, less is known in the deep ocean. Therefore, the role of Phaeodaria in mediating silica through production, export, and dissolution is unique, leaving the fate of deeply produced bSi still not sufficiently assessed.

Due to the porous nature of Phaeodaria skeletons[8], the skeleton of epipelagic populations dissolves in the upper part of the ocean[43]. As a result, dissolved silica is spread throughout the water column, with the potential to resurface through physical mixing, and export to the deep ocean is minimal. In contrast, as bSi production by mesopelagic Phaeodaria occurs in deeper waters, we can expect that their carcasses also reach greater depths before undergoing total dissolution,

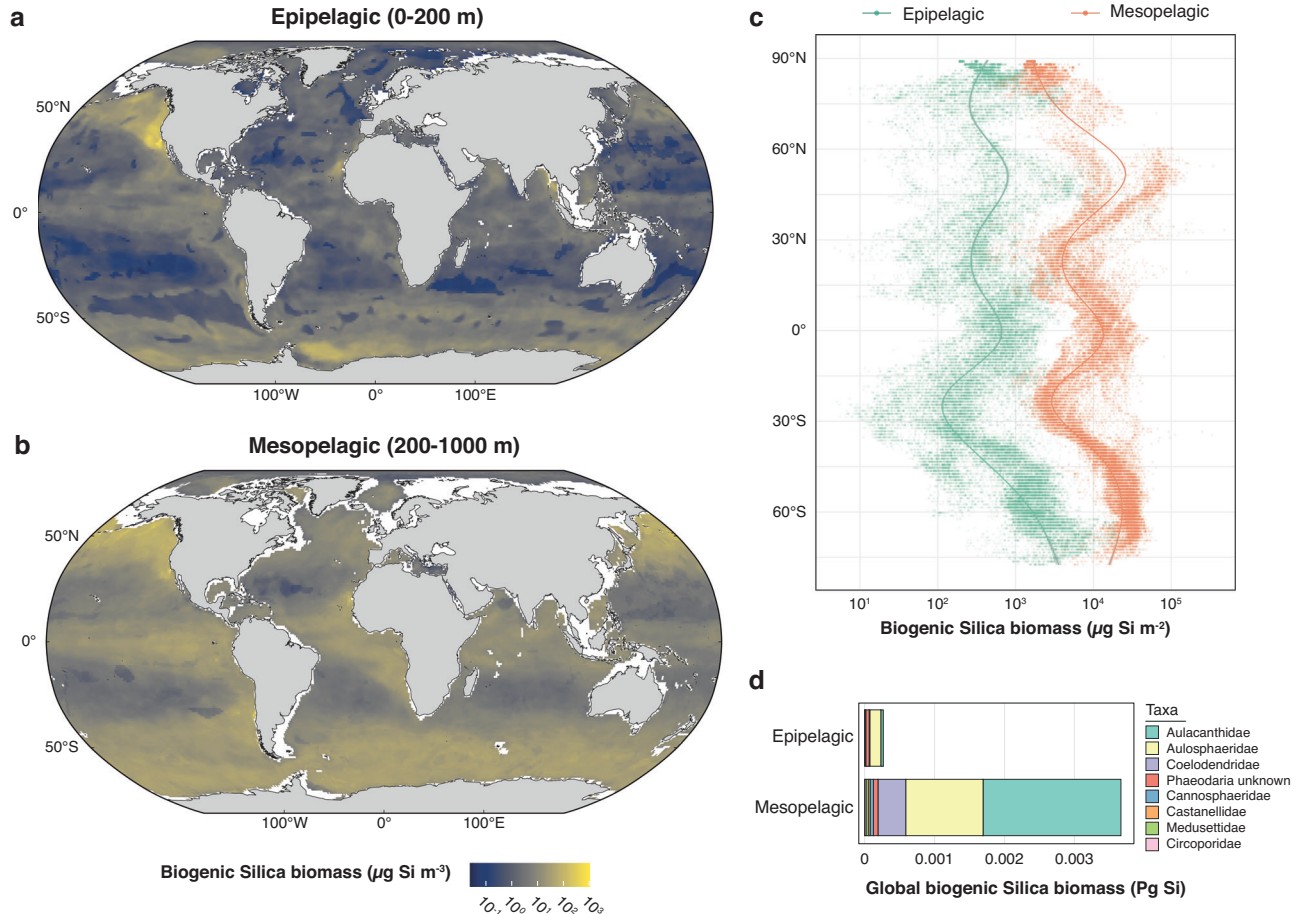

**Fig. 4 | Predicted biogenic silica (bSi) biomass of planktonic Rhizaria. a, b** Maps of the predicted average 1° × 1° biogenic silica concentration in the epipelagic (0–200 m; **a**) and mesopelagic layer (200–1000 m; **b**). **c** Integrated bSi biomass as a function of latitude for both layers. Regression curves were derived using Generalized Additive Models. Note the logarithmic scaling for bSi biomass. **d** Total bSi biomass for Phaeodaria families in both layers. Only groups whose model $R^2$ calculated by random cross-validation is >0.05 are shown. **a, b** Maps were created using the R software version 4.0.3 (ref. 63).

exporting and spreading bSi more efficiently to the deep ocean. Furthermore, as Phaeodaria accumulate siliceous material within their phaeodium, with quantities often reaching the same magnitude as their skeletal structure, it likely enhances the sinking velocity of both the cell and the organic and inorganic materials it carries[44]. Nevertheless, dissolution of bSi before it reaches the seafloor is likely, preventing preservation and fossilization of fragile large Phaeodaria skeletons in deep-sea sediments. The contribution of deep-living Phaeodaria to fluxes, along with the susceptibility of their skeleton to dissolution, has been hypothesized to be an important factor of silica recycling in the North Pacific[45].

Our results suggest that mesopelagic Phaeodaria are important producers and recyclers of bSi, besides organic matter, particularly in the deep ocean. Diatoms have been considered to be the main driver of the silicon cycle globally since the Mesozoic era, when they took over the control from Rhizaria and marine sponges[46]. Our results therefore suggest that Rhizaria, diatoms and sponges still co-dominate the silicon cycle globally, and also provide crucial elements for the future changing oceans. Indeed, diatom populations are thought to decline in the future ocean due to increased water column stratification and thus a decreased nutrient supply into the euphotic zone[47], or a pH-driven decrease in silica dissolution of sinking minerals causing a reduced recycling of silicic acid into the surface ocean[48]. Given the present results, we further explore the consequences of such changes also for Rhizaria populations and their impact on biogeochemical cycles.

## Implications for future ocean biogeochemistry and limitations

Rhizaria have populated the global ocean since 515 Ma and survived all major extinctions[11]. Because of climate change, significant impacts on all oceanic regions are expected, such as seawater warming from the surface to the deep or increased stratification[47]. In future more oligotrophic oceans[26], we can expect mixotrophic Rhizaria populations, thriving in oligotrophic gyres (Supplementary Fig. 3), to remain stable or to expand their habitat range globally. These organisms create microenvironments of enhanced primary production to meet their carbon needs[49]. As such, they will likely not be impacted by changes in primary productivity in surrounding waters and may be favored by elevated temperatures (Supplementary Fig. 5). In addition, they may also profit from a shift towards smaller phytoplanktonic prey, as mixotrophic Radiolaria are known to feed on small organisms[25]. Consequently, their importance on food webs may increase in the future.

In contrast, the fate of Phaeodaria is less certain. These protists are influenced by surface chlorophyll *a* concentrations (Supplementary Fig. 5) and their distribution areas may decrease because of expanding oligotrophic oceans[26]. However, they are generalists, feeding on diverse types of particles[14,15]. Therefore, they may adapt to changes in the upper phytoplankton community composition. Nevertheless, the response of export flux to global changes is highly uncertain as fluxes may increase at high latitudes, while decreasing in equatorial and upwelling areas[4]. Consequently, we can expect Antarctic or subarctic Phaeodaria populations to remain stable, or even become more important. In contrast, upwelling and equatorial populations could decline following such changes. These organisms primarily inhabit the deep ocean, where silica concentrations exceed their requirements and where they are its exclusive users[40]. We can therefore expect them to remain unaffected by changes in the silicon cycle and likely increase their control, leading to more bSi dissolution and recycling at depth. Altogether, their ubiquity as well as the variety of their trophic modes make Rhizaria persistent key organisms also in future oceans.

Our findings advocate for the inclusion of planktonic Rhizaria as a separate compartment in biogeochemical models due to their role in silica and carbon cycling. Yet, in such models, the zooplankton compartment is still inadequately represented. While zooplankton encompass both mixotrophs and heterotrophs, their diversity is often limited to size classes, such as microzooplankton and mesozooplankton[50]. This is partly due to the lack of information about its various components. By refining our knowledge regarding the contrasting distribution and role of mixotrophic and heterotrophic Rhizaria, we provide synoptic information to better represent them in biogeochemical models. However, their impact on element turnover is likely to change, calling repeated assessments of their role in biogeochemical cycles in the future.

Our study provides a comprehensive assessment of the role of giant planktonic Rhizaria in the biogeochemical cycles of carbon and silicon, offering a perspective on their significance for ocean biogeochemistry. They are the sole drivers of biogenic silica production in the mesopelagic layer, and likely throughout the deep ocean, co-dominating the silicon cycle with diatoms and sponges worldwide. Despite relatively low carbon biomasses compared to metazoan zooplankton, they could consume up to 9% of the global gravitational carbon flux worldwide, diminishing the transfer efficiency through the mesopelagic layer. However, their ability to export material through both ballasted dead bodies and mini-pellets may counterbalance this effect. The fate of these substantial biomasses in vertical or silica dissolution fluxes therefore remain to be investigated. However, the present study did not account for smaller, yet more abundant[38], Rhizaria taxa. These may process matter more quickly due to shorter turnover times and could increase recycling of organic matter and bSi throughout the water column. While we expect a limited impact of climate change on overall Rhizaria populations, uncertainties remain regarding the evolution of biogeochemical processes in the mesopelagic zone and the role of Rhizaria in them. Future research should therefore focus on including the full size spectrum of Rhizaria and their multiple roles within mesopelagic food webs and pathways through which they mediate carbon and silica.

## Methods

### Global underwater vision profiler 5 dataset

We used a global UVP5 dataset from 64 oceanographic cruises covering a 13-year period (2008–2021; Supplementary Table 1), which took place across all oceans and across a large range of oceanic structures (Fig. 1a). Among all profiles collected, 4252 covered the first 200 m and 1959 the first 1000 m (Supplementary Table 1). Sampling occurred throughout the year, except at high latitudes where access is limited to boreal or austral summers (Fig. 1b).

The UVP5 images a water volume of -1 L every 5–20 cm of the water column during the descent part of a vertical profile. The onboard computer measures all particles larger than -0.1 mm, but stores vignettes for particles >0.6 mm only[23]. Upon recovery, vertical profiles are processed to extract images, which are associated with a set of metadata and morphological measurements.

Rhizaria images were classified by supervised machine learning algorithms and validated by taxonomy experts on the EcoTaxa web application[51]. In total, 167,551 Rhizaria images were validated, and classified into 18 subgroups belonging to Radiolaria (i.e., Acantharia, Collodaria, and Orodaria), Foraminifera, Phaeodaria and unidentified Rhizaria, following the latest classification for in situ Rhizaria images[12] (Fig. 2 and Supplementary Table 2). Phaeodaria were represented by the two abundant families Aulacanthidae and Aulosphaeridae, and by the less abundant families Cannosphaeridae, Castanellidae, Coelodendridae, Tuscaroridae, plus an additional category "Phaeodaria_unknown". At the lower range of the detection threshold of the UVP5, Acantharia, distinguished by their symmetric spines surrounded by a black center[20], were divided into Acantharia and "Acantharia_like". Collodaria were further classified into colonial specimens, solitary black, and solitary globule[20]. The radiolarian order Orodaria was split between the genus *Cytocladus* and other Orodaria. Foraminifera and other Rhizaria were all classified as such. Our Rhizaria specimens covered a size spectrum ranging from 0.6 mm to 20 mm, the smallest

specimens belonging to Acantharia and the largest to colonial Collodaria (Fig. 2).

## Biomass, carbon demand, and biogenic silica production estimates

For each individual, volume was determined by first computing the Equivalent Spherical Diameter (ESD, in µm) from the surface area of the organism in square pixels converted in µm² (area), including all pixels above a given threshold, extracted by ZooProcess using Eq. (1):

$$ESD = \sqrt{\frac{4 \times area}{\pi}} \quad (1)$$

Then, the volume $V$ (in µm³ cell⁻¹) of the associated sphere was calculated following Eq. (2):

$$V = \frac{4}{3}\pi \times ESD^3 \quad (2)$$

Volumes were also calculated by fitting a prolate ellipse to the object using the major and minor axes length, to ensure that it would not lead to significant differences. Although no significant differences were observed (Supplementary Fig. 6), the sphere method was chosen to estimate volumes as the ellipse method may inflate volumes due to the inclusion of spines in the ellipse.

A volume-to-carbon allometric relationship combining all available carbon content measurements[16,52] was applied to individual volumes $V$ (in µm³ cell⁻¹) to obtain individual carbon contents $Q_C$ (in µg C cell⁻¹) as described by Eq. (3):

$$Q_C = 10^{0.958} \times V^{0.455} \quad (3)$$

For siliceous Phaeodaria, biogenic Si contents $Q_{bSi}$ (in µg Si cell⁻¹) were computed from volumes $V$ (in mm³ cell⁻¹) using the volume-to-biogenic Si allometric relationship reported elsewhere[18] (Eq. 4):

$$Q_{bSi} = 10^{-4.05} \times V^{0.52} \quad (4)$$

Individual carbon demand (CD, in µg C cell⁻¹ d⁻¹) for mesopelagic flux-feeders Phaeodaria was calculated from individual carbon content $Q_C$ following Eq. (5)[22]:

$$CD = \frac{Q_C}{GGE \times \tau_T} \quad (5)$$

with GGE the gross growth efficiency (unitless) and $\tau_T$ the turnover time (in d). Assuming that turnover times are temperature-dependent with a $Q_{10}$ of 2 (i.e., increasing the temperature by 10 °C divides the turnover time by 2), we used a median reference turnover time $\tau$ of 10.9 d at a temperature of 10 °C, calculated over a range of turnover time values estimated between 0 and 150 m[22]. This value was used for all Phaeodaria taxa considered. We also considered the 1st ($\tau = 8.4$ d) and 3rd ($\tau = 17.4$ d) quartiles of the same range of turnover times. Each mesopelagic Phaeodaria specimen was assigned a temperature value coming from the World Ocean Atlas[53] according to its depth, location and month of sampling. The reference turnover time was adjusted to the local temperature using Eq. (6)[20]:

$$\tau_T = \tau \times 2^{\left(\frac{T_{ref} - T_{obs}}{10}\right)} \quad (6)$$

with $T_{ref} = 10$ °C and $T_{obs}$ the observed temperature. GGE, the ratio of prey carbon to predator carbon, ranges between 0.1 and 0.4 and averages 0.2–0.3 for a broad range of protist and metazoan zooplankton[54]. A high value indicates an optimized carbon uptake from food, resulting in minimized energy expenditures. As we expect

Phaeodaria, living in the food-depleted mesopelagic environment, to reduce their energy expenditures, we applied a GGE of 0.4 as reported elsewhere[22] for all considered taxa.

To propose a range of biogenic Si production ($\rho_{Si}$, in µg Si cell⁻¹ d⁻¹), we considered the minimum, maximum, and median from literature values (0.17, 0.54, and 1.78 nmol Si cell⁻¹ d⁻¹)[18] and applied a $Q_{10}$ of 2 as described above for all specimens by Eq. (7):

$$\rho_{Si,T} = \rho_{Si}\left[2^{\left(\frac{T_{ref} - T_{obs}}{10}\right)}\right]^{-1} \quad (7)$$

with $T_{ref}$ and $T_{obs}$ as described above. Daily values are then multiplied by 365 to obtain yearly estimates.

## Environmental data

Temperature, salinity, oxygen and nutrient concentrations (i.e., silicate, phosphate and nitrate) data, from 2008 to 2019, were extracted from the World Ocean Atlas database[53,55–57]. These data were chosen due to their reliance on climatologies calculated on the basis of actual observations, besides their standard use in marine species distribution models (e.g., refs. 58, 59) which enables direct comparability with similar approaches.

They were delineated on a 1° × 1° horizontal grid over a 0–800 m depth range (as silicate, phosphate, and nitrate data were not available deeper) with a monthly temporal resolution covering the years 2008–2019. They were averaged throughout both layer's temporal coverage and depth range. Monthly averaged surface chlorophyll $a$ data, extracted from the Copernicus database, and bathymetric data, extracted from the NOAA database[60], were used for the corresponding time period. These datasets were standardized to a grid resolution of 1° × 1°. UVP5 data were spatially matched to this environmental data on the global 1° × 1° grid. Due to the fact that the World Ocean Atlas does not exactly follow the coastline, profiles that could not be matched to the environmental grid were included into the nearest cell, if any. Finally, calculated individual biomass, C demand, and bSi production were averaged for each layer and each cell[21].

## Predicting global distributions

To model the relationship between environmental variables and Rhizaria distributions, and to ultimately predict global Rhizaria biomass distributions, we used boosted regression trees (BRTs), following a methodology developed recently[21]. Briefly, BRTs function as classical regression trees linking a response (i.e., biomass, carbon demand and biogenic Si production) to predictors (i.e., environmental variables) by performing recursive binary splits[61]. Boosting allows the combination of successive short regression trees, which are adjusted to improve performance on observations poorly modeled by existing trees[62]. They do not produce a single relationship, but instead combine relatively simple successive tree models and are thus well adapted to fit complex, nonlinear relationships between sparse species datasets and their environment, yet being robust against overfitting[62]. BRTs were implemented for each layer and each taxa or group of taxa (i.e., Phaeodaria, each Radiolaria order, Foraminifera, Rhizaria_other, and all Rhizaria). We implemented BRTs using the R software version 4.0.3[63] and the *xgboost* package version 1.2.0.1[64].

To account for spatial correlation between data points, BRTs' performance was tested using random and spatial cross-validation procedures, the latter to improve independence between the training and test sets[65]. For the spatial cross-validation, data were split into five spatial folds according to geographical distances using the R package *blockCV*[66]. Each set consisting of 4 spatial folds was used to train the model, while the remaining fold was used for testing, repeated 20 times. To evaluate the models, we calculated for each test fold (spatial and random) the mean predicted biomass for all repetitions. Then, we calculated the two-sided Pearson correlation coefficient between the

observed biomass and the mean predicted biomass. We calibrated BRTs by trying various combinations of hyperparameters, including the learning rate per tree, the maximum depth of a tree, the minimum number of elements per leaf, and the number of trees, to minimize predictive deviation from the test set (error reduction) and avoid overfitting (complexity minimization)[21,67]. For all models, we chose a learning rate of 0.08, a maximum depth of tree of 2, a minimum number of elements per leaf of 1, and a maximal number of trees of 500. For each layer, we applied models to environmental predictors for all cells of the 1° × 1° world grid, repeated 20 times to obtain maps of the mean and standard deviation of predicted biomass concentrations. We represented the maps of coefficients of variation (standard deviation divided by mean) for the model predicting the carbon biomass of all Rhizaria and the model predicting the bSi biomass of Phaeodaria (Supplementary Fig. 7).

Univariate partial dependence plots were represented to show the effect of each variable on biomass prediction (Supplementary Fig. 5). They were computed by averaging the marginal effect of the variable across model resamples (predicted changes in fitted biomass values for one unit change in the variable). Predicted values were integrated for each layer (in $\mu g\ m^{-2}$) by multiplying the mean concentration (in $\mu g\ m^{-3}$) for this layer by its thickness (in m). Biomass in the top 500 m was estimated by integrating the mean concentration in the epipelagic and the mesopelagic over 300 m. To derive global values, integrated values were multiplied by the area of each 1° × 1° grid cell and then summed over the world as reported elsewhere[21]. To provide an uncertainty range, we derived global values using the mean prediction ± its standard deviation for each cell.

Regional values were obtained by partitioning world predictions using Longhurst's provinces[68]. To estimate the percentage of flux attenuation, we used carbon export (i.e., out of the euphotic zone layer) values from the literature[32–34,69] and computed the ratios of Phaeodaria carbon demand to carbon export globally and for each oceanic region.

### Reporting summary
Further information on research design is available in the Nature Portfolio Reporting Summary linked to this article.

## Data availability
The input and output model data generated in this study have been deposited in the Zenodo database [https://doi.org/10.5281/zenodo.10652050][70]. The environmental data used in this study are available in the World Ocean Atlas database at https://www.ncei.noaa.gov/products/world-ocean-atlas. The surface chlorophyll *a* data are available in the Copernicus database at https://data.marine.copernicus.eu/product/OCEANCOLOUR_GLO_BGC_L4_NRT_009_102. Raw Underwater Vision Profiler 5 image datasets are available in the EcoTaxa database (https://ecotaxa.obs-vlfr.fr/) upon request to project owners.

## Code availability
All scripts used for results presented in this paper are available at https://doi.org/10.5281/zenodo.10652050 (ref. 70).

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

## Acknowledgements

This work was funded by the French "Agence Nationale de la Recherche" project RhiCycle (ANR-19-728 CE01-0006). We thank cruise leaders and participants who helped in the creation of the UVP5 dataset. We are grateful for the ship time provided by the respective institutions and programs. We acknowledge all the scientific programs (in particular the grant NASA OBB #80NSSC17K0568 for the EXPORTS program and NASA OBB #NNX15AE67G for the NAAMES program, AWI_PS99_00, AWI_PS124_13, METEOR/MERIAN core program & EU H2020 Project AtlantOS) involved in data acquisition. We are grateful to the people who sorted images and contributed to build this dataset. We thank Michael Stukel for his advice regarding turnover time calculations and Frédéric Le Moigne for his comments on the manuscript. This work is supported by the graduate school IFSEA that benefits from a France 2030 grant "ANR-21-EXES-0011" operated by the French National Research Agency. RK acknowledges support via a Make Our Planet Great Again grant from the French National Research Agency (ANR) within the Programme d'Investissements d'Avenir #ANR-19-MPGA-0012 and funding from the Heisenberg Programme of the German Science Foundation #KI 1387/5-1. L.S. was supported by the CNRS/Sorbonne University Chair VISION to initiate the global observation. A.R. was funded by the PACES II (Polar Regions and Coasts in a Changing Earth System) program of the Helmholtz Association, the Federal Ministry of Education and Research of Germany (BMBF) through the projects "Changing Arctic Transpolar System" (CATS, grant number 03F0776E) and CATS-Synthesis (grant number 03F0831D), as well as the INSPIRES programme of the Alfred Wegener Institute Helmholtz Centre for Polar and Marine Research (AWI).

## Author contributions

T.B. and M.L. conceptualized, conceived and developed the work with input from R.K., A.L., N.L.M., L.S., A.R. and J.-O.I.; R.K., A.R., L.S. and T.B. contributed to data acquisition; T.B. manually validated all rhizarian images; M.L. modified the models developed by L.D., R.K., T.P. and J.-O.I.; M.L. and T.B. contributed substantially to the data analysis with assistance from J.-O.I.; M.L. and T.B. created all figures; M.L. drafted the manuscript, and all authors contributed substantially to its improvement. All authors approved the final submitted manuscript.

## Competing interests

The authors declare no competing interests.
