## [Peer Review File · Nature Communications]

Global census of the significance of giant mesopelagic protists
to the marine carbon and silicon cyclesEditorial Note: This manuscript has been previously reviewed at another journal that is not operating a transparent peer review scheme. This document only contains reviewer comments and rebuttal letters for versions considered at *Nature Communications*.

REVIEWER COMMENTS

Reviewer #1 (Remarks to the Author):

This manuscript summarizes global distributions of Rhizaria in the world's ocean. A large collection of underwater microscopic images is used to construct estimates of the contribution of these planktonic protists to epipelagic and mesopelagic carbon stocks, concluding that the biomass of Rhizaria has been previously grossly overestimated (by orders of magnitude). The study uses the estimated carbon inventories to calculate implied carbon and Si fluxes mediated by these giant protists.

Although the manuscript presents observations very similar to those previously described in Biard et al. 2016, the major departure from that previous work is the use of new carbon-biovolume conversion factors, derived from a prior study by several of the authors. The use of these new conversion factors substantially lowers previous estimates of Rhizaria biomass in the global ocean. The novelty of the study lies in the methods used to derive biomass, yielding very different results than prior work (that used many of the same data). The study further extends the estimates of biomass to include calculations of carbon demand, particle flux attenuation, and silica production by Rhizaria. To use estimates of biomass for these latter calculations, the study makes numerous assumptions, including protist turnover times, gross growth efficiency, and Si production. The study bases the calculations of these rates on previously published literature values, adjusted to account for temperature.

Major Comments:

1. There is a very long introduction to the manuscript describing everything from the biological carbon pump to the evolutionary history of Rhizaria. There are no findings from this study presented until after lines 107. Consider reducing the introductory materials.

2. In the previous study that is relied on for the biovolume to carbon conversion factors (Laget et al. 2022, L&O), a main conclusion was that Rhizaria “size is weakly correlated with silicon uptake rates”. Its not clear how the present study overcame these weak empirical relationships to compute biogenic Si production rates?

3. Could the mesopelagic biomass distributions reflect spatial variability in sinking particle sizes? Particle export from high latitude regions generally has lower flux attenuation, perhaps due to larger phytoplankton in the epipelagic waters. Spatial variation in productivity and phytoplankton communities in the overlying waters are not discussed as possible controls on the observed Phaeodaria biogeography.

4. Why are the estimates of biomass so different in Figure 3 and Extended Figure 2? In figure 3 the mesopelagic biomass exceeds 10^5 ug C m⁻² while in Extended Figure 2 the biomass never reaches 10^2 ug C m⁻²? Even the sum of biomass from the various taxa in Extended Figure 2 do not appear to yield estimates approaching 10^5 ug C m⁻²?

Minor comments:

1. There should be some mention that the high latitude sampling occurred primarily in the summer/fall (both boreal and austral), which might skew the global distribution data (the elevated biomass in the high latitudes may reflect periods of high productivity and export at the time of sampling).

2. In the pdf version of the file, none of the equations (lines 610, 620, 630) are legible - they all contain “??” instead of the correct variable symbols or letters.

Reviewer #2 (Remarks to the Author):

2nd review of Laget et al. “A global census of the significance of giant mesopelagic protists to the biogeochemical 1 cycles of carbon and silicon”

Thank you for your response. I commend the authors for making their data and code

available. I would highlight though that it was difficult to re-review your paper because you hadn't always put the text you'd changed into the response and hadn't included line numbers in some places (e.g., for the r^2 values). Some further comments:

R2.17. Thanks for adding this paragraph. I think, though, your first sentence is too sweeping. There are many aspects of global pelagic systems that might be little affected by Rhizaria. Can you focus the first sentence on your main findings, i.e., Rhizaria could be important for carbon and silica cycling.

R2.18. My understanding is that the primary reason that diatoms are likely to decline in the future is because increased warming leads to increased stratification, a more stable water column and lower nutrients in the euphotic zone, which leads to lower diatom concentrations. Please either include this mechanism as well or show why it's incorrect.

R2.28. Please remove "expert-calculated" as scientists who've developed all climatologies would argue they were "expert-calculated". It would be stronger if you provide a reference for WOA being used in SDMs, especially if it's the de facto standard.

R2.31. Thank you for being explicit. What you had before was vague.

R2.32. You do not give any information where the r^2 values are? I eventually found them in Extended Data Table 3? Are there more? Please drop the significance level. With so many data it is meaningless, unless you adjusted for temporal and spatial autocorrelation, which I don't think you did. Please comment on why the spatial CVs are so low.

Reviewer #2 (Remarks on code availability):

The URL for the code above should be:

<https://zenodo.org/records/10303404>

Reviewer #3 (Remarks to the Author):

General comments

The authors have responded constructively to referees' comments and made relevant changes to improve the manuscript. The most important points are:

- a better description of the equations, allometric relationships and justification of choices for gross growth efficiency and turnover times
- a better description of the families and orders for the taxonomy of Rhizarians
- a better explanation of the involvement of Rhizarians in the carbon cycle and clarification of the dual contribution of Pheodarians to this cycle
- the addition of a paragraph on the implications of the results for biogeochemical models
- a map showing the coefficient of variation of the prediction models (mean, standard deviation) in the supplementary figures.

Paragraphs have been shortened and sentences that were too general have been removed, which improves the overall quality of the manuscript. It now appears ready for publication.

The code and data are shared in a clear way, with modular and commented scripts described in a README file.

Last specific comments on the Main Document:

I. 141.144: “We apply the most recent allometric volume-to-element content relationships^{16,18} to obtain carbon content of all Rhizaria groups, as well as the silica content for Phaeodaria only (as all Phaeodaria are known as silicifying, while silicified Collodaria cannot be distinguished from naked ones in UVP5 images).”

This sentence is much clearer than before, but could be rephrased without brackets to improve the readability even more.

I. 170. Figure 2: Is it normal that tow Rhizaria images are numbered by i?

I. 330-331: “Ingested material is used for Phaeodaria growth but is ultimately processed into

mini-pellets, ejected in the water column and thought to play an important role in carbon export”

If you find it relevant, it could be interesting to suggest hypothesis at that point on this question: does the presence of Pheodarians in a mesopelagic community accelerate the export of carbon to the depths through fecal pellets production and ballast effect; or does it slow down the flux by intercepting high-velocity sinking particles?

l. 466: the paragraph on the importance of taking into account Rhizarians in biogeochemical models is important, but might be improved by rephrasing (sentences are not well articulated and reading is not smooth).

l. 362: “Univariate partial dependence plots were represented to show the effect of each variable on biomass prediction”

Maybe the Extended Data Figure 5 should be cited?

Reviewer #3 (Remarks on code availability):

I was able to download the code and open the scripts. The data, figures and results are available in folders with a clear organisation. Scripts are modular and commented, and there is a ReadMe document that explains how to run the code, making this work transparent and reproducible.

Reviewer #1 (Remarks to the Author):

This manuscript summarizes global distributions of Rhizaria in the world's ocean. A large collection of underwater microscopic images is used to construct estimates of the contribution of these planktonic protists to epipelagic and mesopelagic carbon stocks, concluding that the biomass of Rhizaria has been previously grossly overestimated (by orders of magnitude). The study uses the estimated carbon inventories to calculate implied carbon and Si fluxes mediated by these giant protists.

Although the manuscript presents observations very similar to those previously described in Biard et al. 2016, the major departure from that previous work is the use of new carbon-biovolume conversion factors, derived from a prior study by several of the authors. The use of these new conversion factors substantially lowers previous estimates of Rhizaria biomass in the global ocean. The novelty of the study lies in the methods used to derive biomass, yielding very different results than prior work (that used many of the same data). The study further extends the estimates of biomass to include calculations of carbon demand, particle flux attenuation, and silica production by Rhizaria. To use estimates of biomass for these latter calculations, the study makes numerous assumptions, including protist turnover times, gross growth efficiency, and Si production. The study bases the calculations of these rates on previously published literature values, adjusted to account for temperature.

Major Comments:

1. There is a very long introduction to the manuscript describing everything from the biological carbon pump to the evolutionary history of Rhizaria. There are no findings from this study presented until after lines 107. Consider reducing the introductory materials.

We reduced the introductory material by shortening some sentences and paragraphs. However, to comply with the formatting style of Nature Communications, we added a last paragraph in the Introduction containing a brief summary of our main results.

2. In the previous study that is relied on for the biovolume to carbon conversion factors (Laget et al. 2022, L&O), a main conclusion was that Rhizaria "size is weakly correlated with silicon uptake rates". Its not clear how the present study overcame these weak empirical relationships to compute biogenic Si production rates?

In the manuscript, we did not use relationships but median and quartile values from bSi production from the literature. This is specified in the Method section (lines 686-691):

“To propose a range of biogenic Si production (pSi , in $\mu g Si cell^{-1} d^{-1}$), we considered the minimum, maximum and median from literature values (0.17, 0.54 and 1.78 $nmol Si cell^{-1} d^{-1}$)¹⁸ and applied a Q10 of 2 as described above for all specimens:

$SiT = Si[2(Tref - Tobs10)]^{-1}$

with Tref and Tobs as described above. Daily values are then multiplied by 365 to obtain yearly estimates.”

3. Could the mesopelagic biomass distributions reflect spatial variability in sinking particle sizes? Particle export from high latitude regions generally has lower flux attenuation, perhaps due to larger phytoplankton in the epipelagic waters. Spatial variation in productivity and phytoplankton communities in the overlying waters are not discussed as possible controls on the observed Phaeodaria biogeography.

According to the literature, Phaeodaria are generalist feeders and can feed upon various detritus types. Indeed, they benefit from an increased particle flux, as specified lines 232-234 (“these non-motile organisms are floating in particle-rich zones, where they intercept aggregates by extending cytoplasmic strands”).

We added the hypothesis that their distribution may be controlled by the presence of large particles by modifying the sentence lines 234-238 from:

“Often thought to be restricted to deep waters⁸, our observations reveal important epipelagic biomasses of Phaeodaria in several high latitude areas, which is in agreement with previous observations in the Southern Ocean^{30–32}, in particular the Weddell Sea³², the North Pacific^{20,33}, but also the Sea of Japan³⁴.”

to

“Often thought to be restricted to deep waters⁸, our observations reveal important epipelagic biomasses of Phaeodaria in several high latitude areas, where they could feed on the sinking of large particles. This is in agreement with previous observations in the Southern Ocean^{30–32}, in particular the Weddell Sea³², the North Pacific^{20,33}, but also the Sea of Japan³⁴.”

We further discussed their adaptation to community composition at lines 383-387: “These protists are influenced by surface chlorophyll a concentration (Supplementary Fig. 5) and their distribution areas may decrease because of expanding oligotrophic oceans²⁹. However, they are generalists, feeding on diverse types of particles^{14,15}. Therefore, they may adapt to changes in upper phytoplankton community composition.”

4. Why are the estimates of biomass so different in Figure 3 and Extended Figure 2? In figure 3 the mesopelagic biomass exceeds $10^5 \mu g C m^{-2}$ while in Extended Figure 2 the biomass never reaches $10^2 \mu g C m^{-2}$? Even the sum of biomass from the various taxa in Extended Figure 2 do not appear to yield estimates approaching $10^5 \mu g C m^{-2}$?

This was a mistake regarding units, the Extended Figure 2 was in mg instead of μg as specified. Thank you for pointing this out, the scale in Extended Figure 2 is now corrected.

Minor comments:

1. There should be some mention that the high latitude sampling occurred primarily in the summer/fall (both boreal and austral), which might skew the global distribution data (the elevated biomass in the high latitudes may reflect periods of high productivity and export at the time of sampling).

We added the sentence in the main text “However, it should be kept in mind that high latitude sampling mostly occurred in boreal and austral summers, likely during periods of highest biomass.” lines 189-191.

We also mentioned this point in the Method section, lines 621-622: “Sampling occurred throughout the year, except at high latitudes where access is limited to boreal or austral summers (Fig. 1b).”

2. In the pdf version of the file, none of the equations (lines 610, 620, 630) are legible - they all contain “??” instead of the correct variable symbols or letters.

We apologize for this and made sure that symbols appear in the revised manuscript.

Reviewer #2 (Remarks to the Author):

2nd review of Laget et al. “A global census of the significance of giant mesopelagic protists to the biogeochemical 1 cycles of carbon and silicon”

Thank you for your response. I commend the authors for making their data and code available. I would highlight though that it was difficult to re-review your paper because you hadn't always put the text you'd changed into the response and hadn't included line numbers in some places (e.g., for the r^2 values). Some further comments:

We apologize for this. We made our data and code available at <https://doi.org/10.5281/zenodo.10652050> as specified in the “Data availability statement” lines 428-435 and “Code availability statement” lines 437-440.

We also refer to the R^2 values in the main text, as specified below.

R2.17. Thanks for adding this paragraph. I think, though, your first sentence is too sweeping. There are many aspects of global pelagic systems that might be little affected by Rhizaria. Can you focus the first sentence on your main findings, i.e., Rhizaria could be important for carbon and silica cycling.

We changed the first sentence of this paragraph lines 397-400 from:

“It is crucial to include Rhizaria in biogeochemical models for accurately understanding and predicting the future of pelagic ecosystems.”

to

“Our findings advocate for the inclusion of planktonic Rhizaria as a separate compartment in biogeochemical models due to their role in silica and carbon cycling.”

R2.18. My understanding is that the primary reason that diatoms are likely to decline in the future is because increased warming leads to increased stratification, a more stable water column and lower nutrients in the euphotic zone, which leads to lower diatom concentrations. Please either include this mechanism as well or show why it's incorrect.

The sentence lines 363-366 was modified from:

“Indeed, a pH-driven decrease in silica dissolution of sinking minerals is likely to lead to a subsequent decline in diatom populations due to a reduced availability of silicic acid in the surface ocean.”

to

“Indeed, diatom populations could decline because of increased water column stratification and lower nutrient supply to the surface ocean⁴⁷, or because of reduced availability of silicic acid in the surface ocean due to a pH-driven decrease in silica dissolution of sinking minerals⁴⁸.”

R2.28. Please remove “expert-calculated” as scientists who've developed all climatologies would argue they were “expert-calculated”. It would be stronger if you provide a reference for WOA being used in SDMs, especially if it's the de facto standard.

We provided two example references which use WOA in SDMs, for phytoplankton as well as for larger organisms, and removed “expert -calculated”.

We modified the sentence lines 696-698 from:

“These data were chosen due to their reliance on expert-calculated climatologies on the basis of actual observations, besides their standard use in marine species distribution models which enable direct comparability with similar approaches.”

to

“These data were chosen due to their reliance on climatologies calculated on the basis of actual observations, besides their standard use in marine species distribution models (e.g., ref.58,59) which enables direct comparability with similar approaches.”

R2.31. Thank you for being explicit. What you had before was vague.

R2.32. You do not give any information where the r^2 values are? I eventually found them in Extended Data Table 3? Are there more? Please drop the significance level. With so many data it is meaningless, unless you adjusted for temporal and spatial autocorrelation, which I don't think you did. Please comment on why the spatial CVs are so low.

We removed the significance level in Supplementary Tables 2 and 3.

To introduce spatial cross-validation in the main text and specify the location of R2 values, we modified the end of the first subsection of the Results (Modeling Rhizaria distribution) lines 139-146 from:

“Associating each of the 167,551 Rhizaria images with a set of environmental variables, we quantify these processes on a global 1°×1° grid by using boosted regression trees (see Methods)” to

“Associating each of the 167,551 Rhizaria images with a set of environmental variables, we quantify these processes on a global 1°×1° grid by using boosted regression trees, with both random and spatial cross-validation (CV) for model assessment. We used spatial CV to account for the fact that spatial data are not independent and to assess the performance of our models on new environmental conditions. For each model and CV method, we provide a R2 value (Supplementary Table 2, Supplementary Table 3), calculated as the squared Pearson correlation coefficient between the observed biomass and the mean predicted biomass (see Methods).”

We comment on why the spatial CVs are so low in the 2nd subsection of the Results (Global Rhizaria biomass from the epipelagic to the mesopelagic) by adding the following lines 174-180: “Yet, large discrepancies can be observed in model assessment (R2 metrics; Supporting Table 2) depending on the use of random or spatial CV. By using a spatial CV, we purposely exclude part of observed environmental conditions, leading to highly conservative R2 values. Low observed R2 values when using spatial CV show the difficulty for our models to predict on new data, but it should be kept in mind that the final model is trained on the entire dataset and therefore includes a larger range of environmental conditions.”

Reviewer #2 (Remarks on code availability):

The URL for the code above should be:

<https://zenodo.org/records/10303404>

The new version of the code is provided in the Code availability statement lines 442-445.

Reviewer #3 (Remarks to the Author):

General comments

The authors have responded constructively to referees' comments and made relevant changes to improve the manuscript. The most important points are:

- a better description of the equations, allometric relationships and justification of choices for gross growth efficiency and turnover times
- a better description of the families and orders for the taxonomy of Rhizarians
- a better explanation of the involvement of Rhizarians in the carbon cycle and clarification of the dual contribution of Pheodarians to this cycle
- the addition of a paragraph on the implications of the results for biogeochemical models
- a map showing the coefficient of variation of the prediction models (mean, standard deviation) in the supplementary figures.

Paragraphs have been shortened and sentences that were too general have been removed, which improves the overall quality of the manuscript. It now appears ready for publication.

The code and data are shared in a clear way, with modular and commented scripts described in a README file.

We thank the reviewer for her/his positive comment regarding our revised manuscript.

Last specific comments on the Main Document:

I. 141.144: “We apply the most recent allometric volume-to-element content relationships^{16,18} to obtain carbon content of all Rhizaria groups, as well as the silica content for Phaeodaria only (as all Phaeodaria are known as silicifying, while silicified Collodaria cannot be distinguished from naked ones in UVP5 images).”

This sentence is much clearer than before, but could be rephrased without brackets to improve the readability even more.

We divided the sentences lines 133-137 into two sentences as follow:

“We apply the most recent allometric volume-to-element content relationships^{16,18} to obtain carbon content of all Rhizaria groups, as well as the silica content for Phaeodaria only. Indeed, all Phaeodaria are known as silicifying, while silicified Collodaria cannot be distinguished from naked ones in UVP5 images.”

I. 170. Figure 2: Is it normal that tow Rhizaria images are numbered by i?

Yes. Because Acantharia have two major morphologies which we wanted to highlight in this figure.

I. 330-331: “Ingested material is used for Phaeodaria growth but is ultimately processed into mini-pellets, ejected in the water column and thought to play an important role in carbon export”

If you find it relevant, it could be interesting to suggest hypothesis at that point on this question: does the presence of Pheodarians in a mesopelagic community accelerate the export of carbon to the depths through fecal pellets production and ballast effect; or does it slow down the flux by intercepting high-velocity sinking particles?

We added this hypothesis and rewrote the part lines 291-296 from:

“Indeed, mini-pellet abundance can be almost 5 orders of magnitude higher than that of krill fecal pellets in the Weddell Sea³². Therefore, the nature of the exported material - either fecal pellets or aggregated bodies^{37,38} - and the associated differences in sinking velocities and carbon contents as well as the resulting effects on global export must be further explored.”

to

“Indeed, mini-pellet abundance can be almost 5 orders of magnitude higher than that of krill fecal pellets in the Weddell Sea³². This leaves an open question regarding whether mesopelagic Phaeodaria ultimately slow down or accelerate downward fluxes. To answer it, the nature of the

exported material - either fecal pellets or aggregated bodies^{37,38} - and the associated differences in sinking velocities and carbon contents as well as the resulting effects on global export must be further explored.”

I. 466: the paragraph on the importance of taking into account Rhizarians in biogeochemical models is important, but might be improved by rephrasing (sentences are not well articulated and reading is not smooth).

We rewrote the paragraph lines 397-410 as follows:

“Our findings advocate for the inclusion of planktonic Rhizaria as a separate compartment in biogeochemical models due to their role in silica and carbon cycling. Yet, in such models, the zooplankton compartment is still inadequately represented. While zooplankton encompass both mixotrophs and heterotrophs, their diversity is often limited to size classes, such as microzooplankton and mesozooplankton⁵⁰. This is partly due to the lack of information about its various components. By refining our knowledge regarding the contrasting distribution and role of mixotrophic and heterotrophic Rhizaria, we provide elements to better represent them in biogeochemical models. However, their impact on element turnover is likely to change, calling for repeated assessments of their role in biogeochemical cycles in the future.”

I. 362: “Univariate partial dependence plots were represented to show the effect of each variable on biomass prediction”

Maybe the Extended Data Figure 5 should be cited?

The citation was added.

Reviewer #3 (Remarks on code availability):

I was able to download the code and open the scripts. The data, figures and results are available in folders with a clear organisation. Scripts are modular and commented, and there is a ReadMe document that explains how to run the code, making this work transparent and reproducible.

We thank the reviewer for this positive comment.